# Streamlined Preoperative Iron Deficiency Screening and IV Treatment for Colorectal Cancer Patients beyond Clinical Trials

**DOI:** 10.3390/jcm13196002

**Published:** 2024-10-09

**Authors:** Caroline Erichsen, Victoria Rosberg, Peter-Martin Krarup

**Affiliations:** 1Faculty of Health Sciences, University of Copenhagen, Blegdamsvej 3B, DK-2200 Copenhagen, Denmark; 2Digestive Disease Center, Copenhagen University Hospital, Bispebjerg, Nielsine Nielsens Vej 11, DK-2400 Copenhagen, Denmark

**Keywords:** colorectal cancer, intravenous iron, prehabilitation, iron deficiency anemia

## Abstract

**Background/objectives:** Iron deficiency anemia is common in patients with colorectal cancer and affects postoperative outcomes. Preoperative intravenous iron supplementation corrects anemia effectively; however, the effect on the postoperative clinical course is unclear. The aim of this study was to investigate the effects of implementing a screening program for iron deficiency anemia and correction in patients with colorectal cancer. **Methods:** This was a retrospective single-institutional quality-assurance study that included patients undergoing elective surgery for colorectal cancer between January 2019 and December 2021. On 1 March 2020, screening for iron deficiency was implemented. Anemia was corrected by intravenous ferricarboyxmaltose. Patients with hemoglobin <7mmol/L and ferritin <30 µg/L or ferritin 30–100 µg/L and TSAT < 0.2 were divided into iron- and non-iron groups. The primary outcome was a change in hemoglobin concentration, and secondary outcomes included blood transfusion, complications, length of stay, mortality, and implementation efficacy. Data from the patients were extracted from digital patient charts and entered into a database. **Results:** A total of 532 patients were included, of which 177 patients (33.3%) were anemic, 63 were in the iron group, and 114 were in the non-iron group. Adherence to the screening program was 86.3%. Hemoglobin (iron group) increased from a mean of 5.7 mmol/L (SD 0.8) to 6.9 (0.8) mmol/L, *p* < 0.001. Iron treatment decreased transfusion rates from 27.7% to 9.1%, *p* = 0.007, and increased the rate of patients without complications from 53.2% to 79.6%, *p* < 0.001, which held true after controlling for confounding. In addition, 90-day mortality was lower in the iron group; however, this was not significant. **Conclusions:** Administration of intravenous iron increased hemoglobin, leading to reduced use of blood transfusion and fewer complications.

## 1. Introduction

Anemia is prevalent in approximately 1/3 of patients with colorectal cancer at the time of diagnosis. In most cases, anemia will be caused by iron deficiency [1,2], characterized by low hemoglobin levels, depleted ferritin storage, and low transferrin saturation, either by chronic blood loss directly from the tumor and/or malnutrition with insufficient iron intake or tumor-associated inflammation, as previously reviewed [3]. Previous studies have demonstrated that preoperative iron deficiency anemia is associated with a range of postoperative complications including increased risk of nosocomial infections, increased risk of allogenic blood transfusion, increased 30-day morbidity and mortality, longer hospital stays, and reduced disease-free survival [4]. It is therefore recommended, by the World Health Organization (WHO) and in the Enhanced Recovery After Surgery (ERAS) guidelines, that correction of a low hemoglobin concentration should be carried out prior to surgery [5,6]. In Denmark, the Danish Colorectal Cancer group has recommended preoperative iron deficiency correction [2]. Greater focus on prehabilitation, including correction of anemia, may lead to a better physiological starting point before surgery and thereby faster recovery [7]. Moreover, the risk of blood transfusion increases with anemia and transfusion seems to be associated with increased postoperative infection, decreased postoperative recovery, and increased cancer recurrence [6,8].

Currently, intravenous iron administration is the best method to correct preoperative iron deficiency anemia [9]. Compared with oral iron administration, it is more effective at restoring ferritin and hemoglobin levels because it overcomes the problems of malabsorption [9,10]. A randomized trial from 2016 demonstrated that administration of intravenous iron reduced the rate of blood transfusions and length of stay. In addition, a higher concentration of hemoglobin was observed 4 weeks after surgery compared with the non-iron group [4]. Intravenous administration of the iron solution ferriccarboxymaltose has a low risk of hypersensitivity reactions, known as Fishbane reactions, and is generally better tolerated than oral iron [11].

A caveat is that patients with hematological disorders should not be considered in the abovementioned context. These patients have other reasons for anemia with suppressed cell lines and should be offered balanced transfusion including thrombocyte pools prior to surgery. Intravenous iron may not be sufficient in these cases.

The objective of this study was to investigate changes in perioperative outcomes in patients undergoing routine iron deficiency screening and preoperative correction with intravenous iron compared with patients not screened.

## 2. Materials and Methods

### 2.1. Population and Study Design

The Danish healthcare system is publicly funded and free to all citizens. Patients with colorectal cancer have the right to be treated within 14 days after diagnosis. However, prehabilitation including iron deficiency correction can postpone surgery for another 2 weeks (4 weeks in total from diagnosis).

This was a retrospective single-institutional cohort quality-assurance study performed at Copenhagen University Hospital, Bispebjerg, Copenhagen, Denmark, including all consecutive patients above 18 years with a first-time diagnosis of colorectal cancer undergoing elective surgery at Bispebjerg hospital in the period between January 2019 and December 2021. Patients who underwent emergency surgery or did not have a preoperative hemoglobin test were excluded.

As a tool to identify frail patients, a preoperative screening program for iron deficiency anemia and subsequent preoperative anemia correction with intravenous iron was implemented on 1 March 2020 by the senior author as a standard procedure according to the guidelines of the Danish Colorectal Cancer Group. The previous routine hemoglobin sample was supplemented with samples of serum ferritin and transferrin saturation (TSAT). Iron deficiency was defined as hemoglobin <7.0 mmol/L for both genders, a ferritin concentration < 30 µg/L, or ferritin 30–100 µg/L and TSAT < 0.2 [12]. Implementation was a supplement to the already established preoperative protein supplement for patients with colorectal cancer. There were no dietary restrictions.

Patients who met these criteria for iron deficiency anemia were offered preoperative intravenous iron therapy (screening period), thus creating 2 groups: (A) an iron group and (B) a non-iron group. Patients with severe chronic kidney disease (eGFR < 30 mL/min), ferritin > 800 µg/L, severe liver disease, or contraindications to intravenous iron therapy were not considered for intravenous iron treatment. The iron solution used was ferriccarboxymaltose and administration was aimed to be performed on the same day as the outpatient interview. The dosage was estimated based on the patient’s body weight and hemoglobin concentration using the chart below (Figure 1). Ferriccarboxymaltose was administered at a maximum dosage of 1000 mg. If the required dosage exceeded 1000 mg, a second infusion with the remaining dosage was given one week after the initial administration. The maximum total dose given in the study was 2000 mg. Ferriccarboxymaltose was administered over 30 min, and all patients were observed for 30 min after the end of infusion for any sign of adverse effects.

Surgery was planned to take place 4 weeks after the first iron administration, with a planned control blood sample (hemoglobin, ferritin, and TSAT) the day before surgery.

The primary outcome of this study was the change in hemoglobin concentration four weeks after administration. Secondary outcomes included postoperative complications, blood transfusions, length of hospital stay, and 90-day mortality, as well as adverse events associated with administration.

Data decisive for primary and secondary outcomes, including blood test results, the dose of administered iron, postoperative complications, blood transfusions, length of hospital stay, and 90-day mortality, were retrieved, as well as variables with a potential influence on treatment effect and surgical outcome. These included age, gender, body weight, American Society of Anesthesiologists (ASA) class, WHO Performance Score, and TNM stage. TNM stage was defined by the preoperative CT scan and MRI and postoperatively by pathology. Postoperative complications were reported according to the Clavien Dindo classification.

No changes in the surgical approach were considered for patients with iron deficiency anemia and all patients underwent a laparoscopic operation with a colorectal specialist.

### 2.2. Statistics

Categorical data were expressed as whole numbers and percentages and analyzed using the Chi-squared test. Continuous data were tested for normality and presented and analyzed accordingly, as means and standard deviation (Students *t*-test) or median and interquartile range (Mann–Whitney test). Changes in hemoglobin concentration after iron administration were analyzed using the paired *t*-test. Binary secondary outcomes were adjusted for confounding using multi-variable logistic regression analyses, including potential confounders. A *p*-value < 0.05 was considered significant.

## 3. Results

Between January 2019 and December 2021, a total of 622 patients with a first-time diagnosis of colorectal cancer were identified. After the exclusion of patients with emergency presentation and patients without hemoglobin testing, 532 patients were included (Figure 2). A total of 177 (33.3%) patients had a hemoglobin concentration <7.0 mmol/L, with a mean (SD) level of 5.8 (0.8) mmol/L, compared to 8.6 (0.8) mmol/L in the 355 (76.7%) patients with hemoglobin ≥ 7.0. Hemoglobin concentration < 7.0 mmol/L was associated with higher age, ASA score, and performance status, as well as colonic location and higher cT and cN stage (Table 1).

Of the 177 patients with hemoglobin <7.0 mmol/L, 95 (53.7%) were identified during the screening period and 82 (46.3%) prior to screening. A total of 80 (45.2%) patients had iron deficiency, 15 (8.5%) had sufficient ferritin and transferrin saturation, and 82 (46.3%) had incomplete iron status samples. The majority of incomplete testing, 59/82, was in the pre-screening group, as expected (Table 2). This translates into a complete implementation rate of 82 out of 95 (86.3%) patients with anemia during the screening period.

Preoperative ferriccarboxymaltose was administered in 63 out of 177 (35.6%) patients (iron group), 61 out of 95 (64.2%) in the screening group, and two out of 82 (2.4%) in the non-screening group. (Table 2). The index hemoglobin concentration during screening was lower in the iron group, with a mean of 5.7 mmol/L (SD 0.7 mmol/L) compared to 6.1 mmol/L (SD 0.7 mmol/L) in the non-iron group, *p* = 0.007. The characteristics of iron and non-iron patients are shown in Table 3. One patient experienced a Fishbane reaction.

A total of 62 of the 63 patients in the iron group had their control hemoglobin levels analyzed at a median of 25 days (range 2–45 days) after the first administration. One patient entered palliative care. Hemoglobin levels increased to a mean (SD) of 6.9 (0.8) mmol/L, *p* < 0.001, which corresponds to a 21% increase on average. The planned dosage of ferriccarboxymaltose was administered in 45 of 62 patients, whereas 17 patients had a lower dosage as they did not return for the second administration. A total of 12 (19.4%) patients had a poor response (increase in hemoglobin less than 0.4 mmol/L). The index hemoglobin concentration in responders was lower, with a mean of 5.6 mmol/L (SD 0.8) compared to 6.1 (SD 0.6) in the non-responder group, *p* = 0.026.

Of the 177 patients with anemia, 148 (83.6%) patients proceeded to surgical resection, 54 out of 63 (87.1%) in the iron group and 94 out of 114 (82.5) in the non-iron group. The median time from iron administration to surgery was 28.5 days (IQR 21.75–33.25 days).

Amongst patients undergoing resection, the rate of blood transfusion was lower, 5/54 (9.3%) in the iron group compared with 26/94 (27.7%) in the non-iron group, *p* = 0.008. The median length of hospital stay was similar at 3.5 days in the iron group and 4 days in the non-iron group. In the iron group, 43/54 (79.6%) did not have any postoperative complications (equivalent to Clavien Dindo = 0) compared with 50/94 (53.2%) in the non-iron group, *p* < 0.001. The overall 90-day mortality was 2/54 (3.7%) and 10/148 (6.7%) in the iron and non-iron groups, respectively, *p* = 0.262. After controlling for confounding, iron treatment remained significantly associated with a lower rate of blood transfusion, (OR = 0.28 (0.09–0.86), *p* = 0.026), and a higher rate of hospitalization without complications (OR = 3.95 (1.65–9.46), *p* =0.002). In addition, 90-day mortality (OR = 0.44 (0.08–2.42), *p* = 0.346) was unaltered (Table 4).

## 4. Discussion

In the present study, a third of the patients with a new onset diagnosis of colorectal cancer had anemia based on the set limit of hemoglobin < 7.0 mmol/L, which corresponds with previous studies [1]. As in other studies, patients with right-sided tumors had a higher incidence of iron deficiency anemia compared with left-sided or rectal tumors. This may be due to the fact that right-sided tumors often are diagnosed later than left-sided tumors and at higher stages.

Implementation of iron deficiency screening outside a study protocol proved to be feasible but was only completed in around 80% of the patients with anemia. Reasons for this relatively low adherence were not directly studied; however, some of the patients were referred with a recent hemoglobin result, which was not supplemented. Likewise, only three-quarters of patients with proven iron deficiency anemia received preoperative iron correction. A reason for this could be the physician’s attitude towards the concept of postponing the operation due to anemia correction in patients without critical anemia, which is supported by the fact that the hemoglobin concentration was higher in the group of patients not receiving iron treatment. In addition, patients could decline the treatment. Few differences between the groups were also observed. Patients in the iron group had a slightly lower hemoglobin level and a higher ASA score whereas patients in the non-iron group had a higher tumor stage. These factors may also partly explain the choice of omitting preoperative iron treatment.

In the patients that did receive iron treatment, a significant increase in hemoglobin of more than 1 mmol/L on average measured after 4 weeks was observed. This is comparable with several previous studies demonstrating an increase in hemoglobin of 0.93–1.2 mmol/L [8,13,14], whereas a randomized clinical trial (RCT) demonstrated an increase of only 0.5 mmol/L [4]. There are several differences between this RCT and our study design. Firstly, they used the WHO’s definition of anemia (Hgb < 13 g/dL for men and <12 g/dL for women). These hemoglobin levels are significantly higher compared to the present study (Hgb < 11.28 g/dL (7mmol/L)). It has been shown that patients with lower hemoglobin and ferritin concentrations have the highest increase in hemoglobin levels after intravenous administration of iron [9,12]. Our results support these findings, demonstrated by a higher index hemoglobin concentration in non-responders compared with responders. The optimal hemoglobin cutoff for indication of clinically sound preoperative iron correction has not yet been established, but it is likely not higher than 7 mmol/L, as used here. The time between administration and surgery is also very important. In the previously mentioned RCT, re-testing was performed after 8 days (6–13) compared to 25 days in the present study. An observational cohort study from 2021 examined the hemoglobin change at 1 week, 2 weeks, and 4 weeks after initial intravenous iron treatment and found an incremental increase in hemoglobin, peaking at 1.32 mmol/L after week 4 [12].

With the increase in hemoglobin obtained in this study after intravenous iron treatment, a significant reduction in blood transfusions was demonstrated compared with patients not receiving pre-operative iron. The rate of blood transfusions in the non-iron group was 28%, which was higher than expected. In a systematic review, only one in eight studies reporting this outcome showed a significant decrease in transfusion rates [9]. It may be due to the lower anemia threshold used in this study, as a lower hemoglobin concentration will increase the likelihood of transfusion in the non-iron group. Preoperative treatment with iron resulted in an overall lower complication rate according to the Clavien Dindo score. When we looked at the group of patients without any complications, it was almost 30% higher in the iron group. This translated into a reduction in 90-day mortality of 3.7% compared with 8.5% in the non-iron group. However, this result did not reach statistical significance. A possible explanation could be a low statistical power because of few events and thus an increased risk of type II error.

## 5. Conclusions

In conclusion, the implementation of screening for iron deficiency anemia outside a study protocol was feasible and safe with only one mild Fishbane reaction. Intravenous iron administration resulted in significantly higher hemoglobin concentrations of 21% on average and resulted in lower rates of complications and blood transfusions, which per se have the potential to reduce mortality. As such, it is recommended that all patients with colorectal cancer undergo preoperative iron deficiency screening and subsequent correction with intravenous iron. This study was limited by the retrospective design and the use of historical controls, as well as the limited number of patients available. However, a comparison of consecutive patients before and after the implementation of iron deficiency screening without altercations of other factors is a strength.

The effect of preoperative anemia correction was associated with the concentration of hemoglobin, where 7 mmol/L seems appropriate in this study. However, further data are warranted regarding the hemoglobin threshold as some suggest even higher thresholds according to the WHO anemia criteria. A one-time administration solution for patients with higher iron requirements could improve proper dosing.

## Figures and Tables

**Figure 1 jcm-13-06002-f001:**
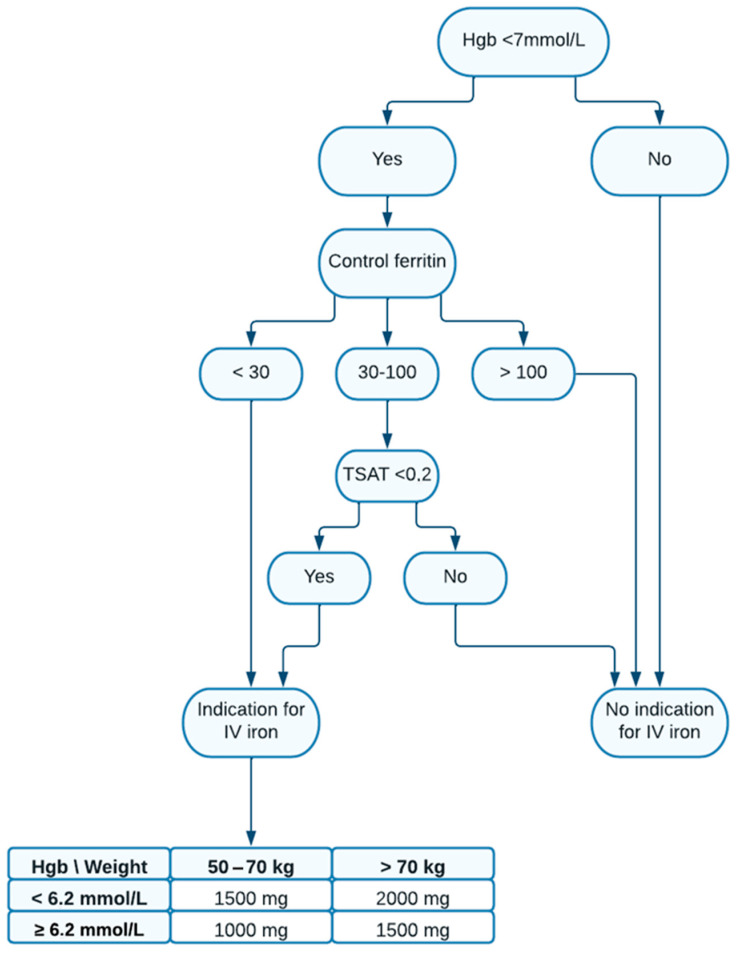
Flowchart showing the criteria used for preoperative IV iron therapy in the intervention group, including the chart for calculation of the iron dose based on the body weight and hemoglobin concentration of the patient.

**Figure 2 jcm-13-06002-f002:**
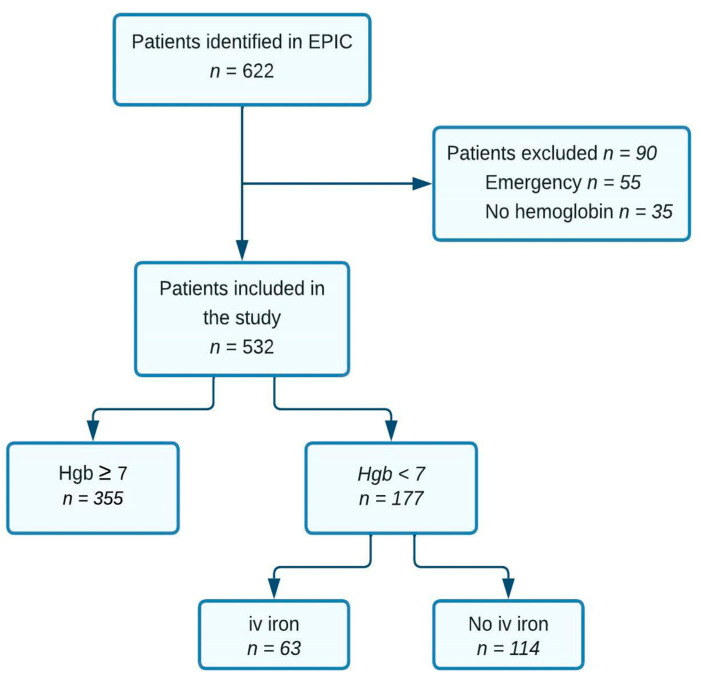
Flowchart of patient inclusion and division into “iron” and “non-iron” groups.

**Table 1 jcm-13-06002-t001:** Patient characteristics of 532 patients with a first-time diagnosis of colorectal cancer.

	Hb ≥ 7.0 mmol/L		Hb < 7.0 mmol/L		*p*
	N = 355	(76.7%)	N = 177	(33.3%)	
Hemoglobin, mmol/L (mean, SD)	8.5	(0.8)	5.8	(0.8)	<0.001
Gender					0.830
Female	173	(48.7%)	88	(49.7%)	
Male	182	(51.3%)	89	(50.3%)	
Age, y (median, IQR)	70	(62–76)	77	(69–83)	<0.001
ASA					<0.001
I	102	(28.7%)	31	(17.5%)	
II	201	(56.6%)	82	(46.3%)	
III	39	(11.0%)	48	(27.1%)	
IV	0	(0.0%)	1	(0.6%)	
Missing	13	(3.7%)	15	(8.4%)	
Performance					<0.001
0	282	(79.4%)	95	(53.7%)	
1	40	(11.3%)	45	(25.4%)	
2	12	(3.4%)	14	(7.9%)	
3	5	(1.4%)	4	(2.3%)	
Missing	16	(4.5%)	19	(10.7%)	
Tumor location					
Colon	240	(67.6%)	160	(90.4%)	<0.001
Rectum	115	(32.4%)	17	(9.6%)	
cT stage					<0.001
1	55	(15.5%)	15	(8.5%)	
2	100	(28.2%)	37	(20.9%)	
3	113	(31.8%)	77	(43.5%)	
4	28	(7.9%)	28	(15.8%)	
Missing	59	(16.6%)	20	(11.3%)	
cN stage					0.006
0	226	(63.7%)	99	(55.9%)	
1–2	75	(21.1%)	58	(32.8%)	
Missing	54	(15.2%)	20	(11.3%)	
Metastatic disease					0.351
No	273	(76.9%)	131	(74.0%)	
Yes	50	(14.1%)	33	(18.6%)	
Missing	32	(9.0%)	13	(7.3%)	
Operation					0.027
None	35	(9.9%)	28	(15.6%)	
Local	15	(4.2%)	1	(0.6%)	
Resection	282	(79.4%)	139	(78.5%)	
Other	23	(6.5%)	9	(5.1%)	
Anemia screening					0.232
No (January 2019–February 2020)	184	(51.8%)	82	(46.3%)	
Yes (March 2020–December 2021)	171	(48.2%)	95	(53.7%)	

**Table 2 jcm-13-06002-t002:** Iron status and treatment of the 177 patients with Hgb < 7 mmol/L. Classified according to whether they are from before or after the implementation of the screening program.

	Screening		No Screening	
**Iron status**	**N = 95**	**(53.7%)**	**N = 82**	**(46.3%)**
Iron deficiency	60	(63.2%)	20	(24.4%)
No iron deficiency	12	(12.6%)	3	(3.7%)
Not assessed	23	(24.2%)	59	(71.9%)
**IV ferriccarboxymaltose**	**N = 61**	**(64.2%)**	**N = 2**	**(2.4%)**
Iron deficiency	46	(75.4%)	0	(0.0%)
No iron deficiency	7	(11.5%)	0	(0.0%)
Not assessed	8	(13.1%)	2	(100%)

**Table 3 jcm-13-06002-t003:** Patient characteristics of 177 patients with hemoglobin levels <7.0 mmol/L.

	i.v. Ferricarboxymaltose		No i.v. Ferricarboxymaltose		*p*
	N = 63	(35.6%)	N = 114	(64.4%)	
Hemoglobin, mmol/L (mean, SD)	5.7	(0.7)	5.9	(0.8)	
Gender					0.297
Female	28	(44.4%)	60	(52.6%)	
Male	35	(55.6%)	54	(47.4%)	
Age, y (median, IQR)	79	(70–83)	77	(68–82)	0.388
Iron deficiency					
No	7	(11.1%)	8	(7.0%)	
Yes	48	(76.2%)	32	(28.1%)	
Not assessed ^1^	8	(12.8%)	74	(64.9%)	
ASA					0.026
I	7	(11.1%)	24	(21%)	
II	29	(46%)	53	(46.5%)	
III	26	(41.3%)	23	(20.2%)	
Missing	1	(1.6%)	14	(12.3%)	
Performance status					0.124
0	32	(50.8%)	63	(55.3%)	
1	20	(31.7%)	25	(21.9%)	
2+	10	(15.9%)	8	(7.0%)	
Missing	1	(1.6%)	18	(15.8%)	
Tumor location					0.299
Colon	55	(87.3%)	105	(92.1%)	
Rectum	8	(12.7%)	9	(7.9%)	
cT stage					0.040
1	9	(14.3%)	6	(5.3%)	
2	18	(28.6%)	19	(16.7%)	
3	23	(36.5%)	54	(47.4%)	
4	9	(14.3%)	19	(16.7%)	
Missing	4	(6.3%)	16	(14.0%)	
cN stage					0.0150
0	41	(65.1%)	58	(50.9%)	
1–2	17	(27.0%)	41	(36.0%)	
Missing	5	(7.9%)	15	(13.2%)	
Metastatic disease	9	(14.3%)	24	(21.1%)	0.429
Surgical procedure					0.901
None	9	(14.3%)	20	(17.5%)	
Right hemicolectomy	32	(50.8%)	58	(50.9%)	
Left hemicolectomy	9	(14.3%)	19	(16.7%)	
Low anterior resection	2	(3.2%)	3	(2.6%)	
Abdominoperineal excision	5	(7.9%)	6	(5.3%)	
Total colectomy	2	(3.2%)	3	(2.6%)	
Other	4	(6.4%)	5	(4.4%)	

^1^ Most of these were prior iron deficiency screening.

**Table 4 jcm-13-06002-t004:** Clinical outcomes in 148 patients with colorectal cancer and hemoglobin < 7.0 mmol/L. ^1^ In surviving patients.

	i.v. Ferricarboxymaltose		No i.v. Ferricarboxymaltose		*p*
	N = 54	(36.5%)	N = 94	(63.5%)	
Blood transfusion	5/54	(9.3%)	26/94	(27.7%)	0.007
Length of stay (median, IQR) ^1^	3.5	(3.0–6.0)	4.0	(3.0–8.0)	0.486
Clavien Dindo (30 days)					0.045
0	43	(79.6%)	50	(55.6%)	
1	2	(3.7%)	14	(15.6%)	
2	3	(5.6%)	13	(14.4%)	
3	3	(5.6%)	4	(4.4%)	
4	1	(1.8%)	5	(5.6%)	
5	2	(3.7%)	4	(4.4%)	
No complications	43/54	(79.6%)	50/94	(53.2%)	<0.001
90-day mortality	2/54	(3.6%)	8/94	(8.5%)	0.251

## Data Availability

Due to the nature of the research and due to ethical and legal restrictions, data are not available.

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
