# Peer review of "Streamlined Preoperative Iron Deficiency Screening and IV Treatment for Colorectal Cancer Patients beyond Clinical Trials"

_jcm, 2024, doi:10.3390/jcm13196002_

Round 1

Reviewer 1 Report

Comments and Suggestions for Authors

The paper is very interesting and very important for clinical practice because it  permit to underline the correction of Iron Deficiency Anaemi before surgical procedure for colon cancer improving the long term survival and avoiding complications. Although, I suggest: 

At page 2, line 42, the Authors could insert the median number of blood transfusions that occur before to start surgical procedure for Colon Cancer

At page 2, the Authors should speak about the management of patients with Colon Cancer and Hematological Disorders different from Iron Deficiency Anaemia (IDA)

At page 11 table 3 the Authors should explain the high percentage of right than left hemicolectomy

Author Response

Thank you taking the time to review our paper and for the comments.

We have added a brief paragraph on hematologic disorders and why this study may not apply for them in the introduction as suggested.

In the Discussion, we have added a section on right vs. left

Howvever I'm not sure what you mean by the median no. of transfusions is that in previous studies of in the present one. Therefore we have not included the information, but are willing to do so if wanted.

Thank you again 

Reviewer 2 Report

Comments and Suggestions for Authors

The study about Iron Deficiency Screening and IV Treatment for Colorectal Cancer Patients is an interesting study. The methods used are appropriate and results are discussed well. To the best of my knowledge, I am suggesting below points-

1.       What’s the novelty of the present work as similar reports already been published in recent years. Clearly spell out the aim of the present work and its novelty.

2.       As it is evident that hemoglobin levels also gets effected by the person’s diet. So during study, was there any diet restriction for the patients?

3.       Modify keywords.

4.       Whats the basis of conducting ‘preoperative intravenous iron therapy’. Is there any legal guidelines or permission which took before this study?

5.       As iron therapy may also be potentially harmful in respect to stimulation of tumour growth, so whats the preventive methods taken to manage this.
Please mention significance level in statistics section.

6.       Line 47, cite the reference- https://doi.org/10.1016/j.phymed.2017.04.010

7.       Define the future prospects of the present work.

Author Response

Thank you for taking the time to review our paper, we appreciate it a lot. 

1) We have clarified the novelty under the aims paragraph in the introduction

2) The following has been added on page 2 "Implementation was a supplement to the already established preoperative protein supplement for patients with colorectal cancer. There were no dietary restrictions."

3) has been modified

4) In Denmark the Danish Colorectal Cancer Group has recommended preoperative iron deficiency correction prior to surgery - this has now been referenced in the introduction. No other legal requirements.

5) The potential risk of tumor growth is only theoretical with no clinical data on colorectal cancer to back it up. However, colorectal cancer is a slow developing cancer and the tumor is resected within 4 weeks, which we, as well as the Danish Colorectal Cancer Group translated into a negligible risk. So no measures were taken. We believe, that operating on anemic patients is more harmful. 

6) the suggested reference is "A phenolic glycoside from Flacourtia indica induces heme mediated oxidative stress in Plasmodium falciparum and attenuates malaria pathogenesis in mice" I don't think that is relevant. 

7) this has been added to the conclusion paragraph under discussion

Thank you again